# Adenosine and Cordycepin Accelerate Tissue Remodeling Process through Adenosine Receptor Mediated Wnt/β-Catenin Pathway Stimulation by Regulating GSK3b Activity

**DOI:** 10.3390/ijms22115571

**Published:** 2021-05-25

**Authors:** Jaeyoon Kim, Jae Young Shin, Yun-Ho Choi, So Young Lee, Mu Hyun Jin, Chang Deok Kim, Nae-Gyu Kang, Sanghwa Lee

**Affiliations:** 1LG Household & Health Care (LG H & H) R&D Center, 70, Magokjoongang 10-ro, Gangseo-gu, Seoul 07795, Korea; kjy5281@lghnh.com (J.K.); sjy2811@lghnh.com (J.Y.S.); youknow@lghnh.com (Y.-H.C.); soyounglee@lghnh.com (S.Y.L.); mhjin@lghnh.com (M.H.J.); ngkang@lghnh.com (N.-G.K.); 2Department of Dermatology, School of Medicine, Chungnam National University, 266, Munwha-ro, Jung-gu, Deajeon 35015, Korea; cdkimd@cnu.ac.kr

**Keywords:** adenosine, cordycepin, wound healing, Wnt/β-catenin pathway

## Abstract

Adenosine is a cellular metabolite with diverse derivatives that possesses a wide range of physiological roles. We investigated the molecular mechanisms of adenosine and cordycepin for their promoting effects in wound-healing process. The mitochondrial energy metabolism and cell proliferation markers, cAMP responsive element binding protein 1 (CREB1) and Ki67, were enhanced by adenosine and cordycepin in cultured dermal fibroblasts. Adenosine and cordycepin stimulated adenosine receptor signaling via elevated cAMP. The phosphorylation of mitogen-activated protein kinase kinase (MEK) 1/2, mammalian target of rapamycin (mTOR) and glycogen synthase kinase 3 beta (Gsk3b) and Wnt target genes such as bone morphogenetic protein (BMP) 2/4 and lymphoid enhancer binding factor (Lef) 1 were activated. The enhanced gene expression by adenosine and cordycepin was abrogated by adenosine A_2A_ and A_2B_ receptor inhibitors, ZM241385 and PSH603, and protein kinase A (PKA) inhibitor H89, indicating the involvement of adenosine receptor A_2A_, A_2B_ and PKA. As a result of Wnt/β-catenin pathway activation, the secretion of growth factors such as insulin-like growth factor (IGF)-1 and transforming growth factor beta (TGFβ) 3 was increased, previously reported to facilitate the wound healing process. In addition, in vitro fibroblast migration was also increased, demonstrating their possible roles in facilitating the wound healing process. In conclusion, our data strongly demonstrate that adenosine and cordycepin stimulate the Wnt/β-catenin signaling through the activation of adenosine receptor, possibly promoting the tissue remodeling process and suggest their therapeutic potential for treating skin wounds.

## 1. Introduction

Skin, the outermost organ of the body, not only protects the host against the environmental stresses such as toxins, UV lights, and microorganisms but also prevents dehydration of the organism. The wound injury causes loss of skin integrity and functions. The restoration of tissue integrity and homeostasis is fundamental for the survival of organisms. The repair of skin wounds is complex and dynamic processes involving diverse cellular responses and surrounding micro-environmental interactions. This wound healing process could be classified into three stages; homeostasis and inflammation, proliferation, and remodeling [1]. During each stage, complex networks of multiple cytokines and mediators facilitate cellular communication between fibroblast and other tissue components [2].

Adenosine, a ubiquitous purine nucleoside produced by dephosphorylation of AMP, has essential roles in modulating tissue homeostasis [3]. Especially, extracellular accumulation of adenosine in response to metabolic stress and/or cell damage [4] has been implicated in tissue protection and repair; increasing oxygen/energy supply, countering inflammation, and enhancing angiogenesis [5,6]. The extracellular adenosine directly activates four subtypes of adenosine receptor, A_1_, A_2A_, A_2B_, and A_3_ [7].

The activation of adenosine receptor plays regulatory roles throughout the body, in virtually all organ systems. In the central nervous system, adenosine modulates neurotransmitter release, synaptic plasticity, and neuroprotection [8,9]. In the cardiovascular system, vasoconstriction and vasodilation are also affected by adenosine receptor activation [10,11]. In the skin, extracellular adenosine is correlated with the promotion of wound healing process. In a mouse model, the activation of adenosine A_2A_ receptor accelerated the wound healing process by stimulating angiogenesis [12] and by affecting fibroblast and epithelial cells [13,14].

*Cordyceps militaris*, a medicinal mushroom, is widely used as a traditional herbal medicine in East Asia. The main active constituent of *C. militaris* fruiting bodies is cordycepin, which was firstly extracted from *C. militaris* [15]. Cordycepin (3′-deoxyadenosine) is a natural derivative of adenosine and was reported to possess diverse pharmacological activities, including anti-oxidation [16], anti-inflammation [17,18], and neuroprotection [19]. In addition, cordycepin prevented cell senescence and senescence-associated secretory phenotype in vitro through modulating the AMPK activity [20].

Because of the structural similarity with adenosine, cordycepin likely behaves as an agonist or an antagonist for target pathways of adenosine, especially for adenosine receptors. Cordycepin was reported to activate adenosine A_2A_ receptor for spontaneous alteration behavior in hippocampus [21] and induce apoptosis by activating adenosine A_3_ receptor in bladder cancer cells [22]. However, studies on the specificity, selectivity, and sensitivity of cordycepin for adenosine receptor subtypes in skin cells such as fibroblasts and keratinocytes are limited.

In this study, we investigated the effects of adenosine and its natural derivative, cordycepin, in cultured human dermal fibroblasts. We examined the expression pattern of adenosine receptor subtypes in fibroblasts and their activation by adenosine and cordycepin treatment. The treatment of adenosine and cordycepin increased the mRNA expression levels of CREB1 and Myc and induced elevation of intracellular cAMP concentration. In addition, mitochondrial energy metabolism was markedly enhanced. As a consequence of adenosine receptor activation by adenosine and cordycepin treatment, the Wnt/β-catenin pathway was found to be stimulated in WRHEK293A reporter cell line and the mRNA expression of Wnt target genes such as BMP2/4 and LEF1 was increased in cultured fibroblasts. We have also found that the phosphorylation of cell signal transduction elements such as mTOR and Gsk3b were increased by adenosine and cordycepin stimulation. Furthermore, both adenosine and cordycepin enhanced the cell migration rate of fibroblast cells in cell scratch assays and stimulated the secretion of growth factors, such as IGF-1 and TGFβ3, reported to be important in the wound healing process. Our data demonstrate that adenosine and cordycepin stimulate Wnt/β-catenin pathway through adenosine receptor activation, potentially promoting the tissue remodeling process in wound recovery and suggest that GSK3b plays a crucial role in interconnecting the adenosine receptor and Wnt signaling pathways.

## 2. Results

### 2.1. Adenosine Receptor A_2B_ Is a Dominant Subtype in Human Dermal Fibroblasts

The cordycepin and 5′-N-Ethylcarboxamidoadenosine (NECA) are derivatives of adenosine, a key nucleoside modulating the diverse physiological processes (Figure 1a). Adenosine exerts its biological effects through four G protein-coupled adenosine receptor subtypes, A_1_, A_2A_, A_2B_, and A_3_ [23]. The selectivity of adenosine and NECA to each adenosine receptors was previously reported [7,24]. Before investigating the effects of adenosine and its derivatives on fibroblast, we analyzed the mRNA expression levels of adenosine receptor subtypes, ADORA1, ADORA2A, ADORA2B, and ADORA3 in cultured human dermal fibroblasts. It was revealed that the adenosine receptor A_2B_ showed the highest expression level among subtypes, approximately 13-fold higher than ADORA1A and ADORA2A. On the contrary, the mRNA expression of the A_3_ subtype was not detected in cultured human fibroblast (Figure 1b). Therefore, it could be presumed that most of the physiological responses triggered by adenosine and its derivatives in cultured human dermal fibroblasts are mediated through the activation of the adenosine receptor A_2B_ subtype.

### 2.2. Changes in mRNA Expression Profiles by Adenosine, Cordycepin, NECA, and Wnt3a in Cultured Human Dermal Fibroblasts

The fibroblast plays a crucial role in the wound healing process [25] and this process is strongly connected with cell development, cell cycle, and growth factor stimulation [26]. Fibroblast cells were treated with adenosine and its derivatives and the mRNA expression levels of 132 genes with diverse functional categories were measured. Recombinant Wnt3a was used as a positive control for Wnt/β-catenin signal activation (Figure 2a). Sixteen genes showed up more than a two-fold increase with statistical significance (*p* < 0.05) and were selected for further investigation, namely protein clustering analysis [27] and pathway estimation following GO and KEGG analysis (Figure 2b). As a result, adenosine and cordycepin were found to activate the pathways related to cellular metabolism and developmental processes (Table 1). Among those processes, Hippo and Wnt signaling pathways were highly predicted to be regulated by adenosine and its derivatives (Table 2). These estimations, based on in-silico modeling, were highly matched with the activation of Wnt signaling pathway by adenosine and cordycepin.

### 2.3. Cordycepin as Well as Adenosine Activated Adenosine Receptor Signaling Pathway in Cultured Human Dermal Fibroblasts

Activation of adenosine receptor induces diverse genes and pathways. We have identified the adenosine receptor A_2B_ subtype as a dominant in fibroblasts (Figure 1b), a G-protein coupled receptor transducing signals in a cAMP dependent manner. The effects of adenosine, cordycepin, and NECA on the intracellular cAMP level were investigated. It was revealed that the cAMP level was increased by adenosine, NECA, and cordycepin treatment within minutes (Figure 3a).

One of the cAMP mediated pathways is energy metabolism [28]. The correlation between cAMP level and enhancement of mitochondrial energy metabolism has been well-established in previous reports [29,30,31]. Both adenosine and cordycepin enhanced the mitochondrial membrane potential (ΔΨ) in cultured fibroblasts. We also have found that the treatment of *Cordyceps Militaris* extract (CME) increased the mitochondrial membrane potential in a concentration dependent manner (Figure 3b). CME was revealed to contain adenosine and cordycepin as major constituents (Appendix A).

The effects of adenosine and cordycepin on genes associated with cAMP and cAMP mediated pathways, such as CREB1 and Myc, were evaluated. The mRNA expression levels of CREB1, Gsk3b, and Myc were increased by both adenosine and cordycepin treatment. In addition, Ki67 and CCND1, the cell cycle progression markers, were increased in concentration dependent manners, more prominent in two-day treatment (Figure 3d). Our data demonstrate that both adenosine and cordycepin stimulate the adenosine receptor pathway in fibroblasts, activating cAMP, cAMP responding genes, and mitochondrial energy metabolism.

### 2.4. Adenosine, Cordycepin and NECA Stimulated Wnt Reporter Activity

The Wnt/β-catenin pathway is one of the critical signaling pathways for the tissue regeneration process [32]. To figure out the possible involvement of adenosine (derivatives) in the wound healing process, the effects of adenosine and its derivatives on the Wnt/β-catenin pathway were investigated in a stable Wnt reporter cell line.

The treatment of adenosine increased luciferase activity in a concentration dependent manner, indicating that adenosine acted as an agonist for Wnt signaling pathway. NECA, one of the major adenosine receptor agonists, also increased the reporter activity. Furthermore, cordycepin (3′-deoxyadenosine) stimulated Wnt/β-catenin signaling in a concentration dependent manner (Figure 4). Accordingly, the stimulation of Wnt signaling by CME treatment was likely caused by adenosine and cordycepin in CME (Figure 4). Our data demonstrate that not only adenosine but also cordycepin and NECA, natural and artificial derivatives of adenosine, stimulate the Wnt signaling pathway.

### 2.5. Adenosine and Cordycepin Activated Wnt/β-catenin Signaling Pathway in Cultured Human Dermal Fibroblasts

As shown in Figure 2, the Wnt/β-catenin signaling related genes were activated by adenosine and cordycepin. In this respect, the mRNA expression of the genes involved in Wnt signaling pathway were further examined in fibroblasts treated with adenosine and cordycepin for one and two days. As target genes of the Wnt signaling pathway, the mRNA levels of BMP2 and BMP4 were elevated by both adenosine and cordycepin. The expression of FZD3 and FZD5, the Wnt receptors, was also increased. The levels of Lef1 and CTNNB1, key transcription factors for the Wnt signaling pathway, were increased by adenosine and cordycepin, more prominently when treated for two days (Figure 5). Our data demonstrate that the activation of the adenosine receptor is closely associated with Wnt/β-catenin signaling in cultured human dermal fibroblasts.

To elucidate the underlying mechanism by which adenosine and cordycepin activate the Wnt signaling pathway, the phosphorylation of MAP kinases was examined by dot blot assay. It was found that the phosphorylation of MEK1/2, mTOR, and p70S6K was significantly increased in adenosine and cordycepin treated groups (Figure 6a,b). In particular, the phosphorylation of Ser9 site of GSK3b was markedly increased, implying its possible role in Wnt signaling activation. The phosphorylation of Gsk3b on S9 site was reported to inactivate the kinase activity, resulting in stimulation of the Wnt signaling pathway. Therefore, we further confirmed the increased phosphorylation of Gsk3b on Ser9 site by western blotting. We have found that adenosine and cordycepin significantly increased the phosphorylation of Gsk3b S9 site in cultured fibroblasts, especially at early time point of treatment (Figure 6c,d), suggesting that adenosine and cordycepin modulated Gsk3b activity via phosphorylation.

Our results demonstrate that adenosine and cordycepin activate both the adenosine receptor and the Wnt/β-catenin signaling pathways in cultured human dermal fibroblasts.

### 2.6. PKA Mediated Gsk3b Inactivation Resulted in Wnt Activation

To investigate the more detailed correlation between the adenosine receptor and the Wnt/β-catenin pathway, the cellular responses to adenosine and cordycepin were examined with several inhibitors. PSH603 [24] and ZM241385 [33] were used for specific inhibitors of the adenosine receptor A_2B_ and A_2A_, respectively. The BMP4 and CREB1 genes were activated by Wnt3a, adenosine, and cordycepin treatment in cultured fibroblasts (Figure 3d and Figure 5). The increased mRNA expression by adenosine and cordycepin treatment was decreased in the presence of PSH603 or ZM241385. The treatment of adenosine receptor inhibitors, however, did not downregulate the mRNA expression induced by recombinant wnt3a treatment in cultured fibroblast (Figure 7a,b). In addition, H89, a PKA inhibitor, reported to stimulate GSK3b activity [34], decreased the mRNA expression levels induced by wnt3a, adenosine, and cordycepin (Figure 7). The results of the inhibitor study were summarized in Table 3.

Our results strongly demonstrate that adenosine and cordycepin stimulate Wnt/β-catenin pathway through the activation of the adenosine receptor and that the inhibitory phosphorylation of Gsk3b by PKA plays a pivotal role in interconnecting the adenosine receptor activation and Wnt signaling in cultured human dermal fibroblasts.

### 2.7. Adenosine and Cordycepin Promoted In Vitro Tissue Repair Process

The tissue repair process requires diverse growth factors for manipulating different skin cell types. The changes in the secretion of growth factors by adenosine and cordycepin were investigated in cultured fibroblasts to elucidate their putative roles in the wound healing process. We have found that the mRNA levels of EGF, IGF-1, TGFB1/2, and VEGFA were increased in concentration dependent manners (Figure 8a). The protein levels of corresponding growth factors were also increased both in cell lysates and culture media (Figure 8c). In particular, the secretion of TGFB3 was most significantly elevated. Therefore, adenosine and cordycepin are expected to increase the supply of growth factors in fibroblasts, critical for wound healing and the skin regeneration process [35].

During the tissue repair process, the dynamics of wound-associated fibroblasts is important for repopulating lost tissue and depositing new matrix, and also contribute to wound closure through the fibroblasts’ contractility. The fibroblasts need to be migrated into a newly formed wound bed [25]. To assess the capability of adenosine and cordycepin to facilitate wound healing, monolayer cell scratch assays were performed. The cell migration rates were found to be significantly increased by adenosine and cordycepin treatment (Figure 9a). In addition, cellular β-tubulin and F-actin were visualized to examine the rearrangement of cytoskeleton that indicate the dynamics of fibroblasts. We have found that the F-actin was drastically increased by adenosine receptor agonists in fibroblasts (Figure 9b), demonstrating that actin filament formation was accelerated by adenosine and cordycepin. Taken together, our data suggest that adenosine and cordycepin could possibly encourage the tissue remodeling process in human dermal fibroblast.

## 3. Discussion

In the present study, we have found that adenosine and cordycepin upregulated the mRNA expression of genes associated with multicellular organismal process and cell proliferation (Table 1), and enhanced cell metabolic processes with increased mitochondrial membrane potential (ΔΨ) (Figure 3c,d). Tissue repair is a high energy consuming process. In this context, the elevated energy metabolism through mitochondrial activation could increase the energy supply needed for wound healing.

The Wnt/β-catenin signaling and Hippo signaling pathways are recognized as critical for development of multi-cellular organisms and tissue regeneration in embryos and adults [36]. In particular, the Wnt/β-catenin signaling pathway strongly contributes to the regeneration of tissues and organs, repairing the internal and external damages in adults [26]. In Figure 4, Figure 5 and Figure 7, the Wnt/β-catenin signaling pathway was stimulated by adenosine and cordycepin treatment via adenosine receptor activation. Diverse phenotypes have been reported as consequences of Wnt/β-catenin signal activation. For instances, the activation of Wnt signaling induces mitochondrial activity in melanoma cells through a PTEN dependent manner [37], contributes to secretion of various growth factors, and encourages cell migration [38]. Our data coincide with previously reported articles in that adenosine and cordycepin stimulated Wnt signaling through the activation of adenosine receptor, increasing mitochondrial activity, growth factor secretion and cell migration in cultured human dermal fibroblasts (Figure 3 and Figure 8).

We have found that the production and secretion of growth factors, such as EGF, IGF-1, TGFβ1/2/3, PDGF and VEGF, were increased by adenosine and cordycepin treatment (Figure 8). EGF and VEGF profoundly contribute to the tissue remodeling through modulating the cell mobility [39]. IGF-1 and TGF β1/2/3 are also essential for the dynamics of macrophage-fibroblast crosstalk in the tissue repair process [40]. In addition, the cooperation between PDGF and IGF signaling synergistically promotes fibroblast proliferation and facilitates wound healing without increased scarring [41]. As shown in Figure 8, adenosine and cordycepin enhanced the production of PDGF and IGF. Our data suggest that co-activation of these signaling pathways could help for the fine-tuning of fibroblasts in the wound healing process.

The TGFβ signaling, when activated by Wnt/β-catenin signaling [42], plays a pivotal role in tissue regeneration with or without scar [43]. In scarless fetal wound healing, the level of TGFβ3 is critical [44]. The scar-free embryonic wounds showed a lower level of TGFβ1 and a higher level of TGFβ3 compared with scar-forming, adult-like wounds in rats [45]. Application of TGFβ3 also resulted in reduced scarring in a rodent wound model [46]. Although some reports were suspicious about the anti-scarring effect of TGFβ3 [47], TGFβ3 definitely affected wound tissue in reducing scar formation through regulation of cell movement and inflammatory mediation [48]. In this study, the secretion levels of TGFβ1 and TGFβ3 were increased by adenosine and cordycepin treatment (Figure 8), showing more marked induction in TGFβ3. It has been reported that the ratio of anti-fibrotic cytokine TGFβ3 to pro-fibrotic cytokine TGFβ1 was higher in fetal skin compared to adult skin [43], which is thought to be one of the key features of scarless wound healing in fetal skin. More precise studies concerning the effects of adenosine and cordycepin on the expression of TGFβ1 and TGFβ3 are needed to elucidate whether they are anti-fibrotic in the tissue remodeling process. Taken together, our data suggest a therapeutic potential of adenosine and cordycepin for treating skin wounds, supplying many growth factors such as TGFβs, VEGF, PDGF and IGF-1, possibly promoting scarless skin repair.

Although we focused on dermal fibroblast cells in the present study, other cell types including keratinocyte and endothelial cells play important roles in the wound healing process and in particular, the Wnt/β-catenin signaling pathway also plays pivotal roles in epidermal regeneration and angiogenesis. Apart from the direct effects of adenosine on keratinocytes or endothelial cells which are unknown yet, several cytokines including EGF, IGF-1, TGFB1/2, and VEGFA stimulated by adenosine might do exert paracrine effects on these cells, for example, epidermal proliferation, wound reepithelization, and angiogenesis. It is not clear, however, adenosine would exert similar effects in these cell types through the same molecular mechanisms since we don’t have any evidences at this moment.

The adenosine A_2A_ and A_2B_ receptor signaling pathways were reported to be involved in wound healing [49]. Our results demonstrate that the cellular signal triggered by the activation of adenosine receptor stimulate the Wnt signaling. As shown in Figure 6, the MAP kinases such as MEK1/2, mTOR and p70S6K were activated by adenosine and cordycepin. In addition, the phosphorylation of Gsk3b, which causes the inactivation of Gsk3b, was increased by adenosine receptor activation (Figure 6c). The phosphorylation of Gsk3b-S9 depends on PKA and PI3K/AKT/mTOR pathways [50] and the Gsk3b-S9 phosphorylation inhibits Gsk3b activity, inhibiting the degradation of β-catenin and activating the Wnt signaling [34]. In Figure 7, the mRNA expression levels of Wnt target genes (BMP4 and CREB1), induced by adenosine and cordycepin treatment, were decreased in the presence of ZM241385 and PSB603, adenosine receptor A_2A_ and A_2B_ inhibitors, respectively. A PKA inhibitor, H89, on the other hand, downregulated the Wnt signaling induced by adenosine and cordycepin treatment, strongly demonstrating the critical role of Gsk3b in interconnecting the activation of Wnt/β-catenin and adenosine receptor pathways.

When adenosine is used as a therapeutics, there could be concerns about adverse effects, especially systemic one since adenosine has profound effects on the heart. We focused on the skin wounds which are mostly external body sites and are the target for topical application. Although the topically applied chemicals could enter the circulation, topical application would be a good regimen to circumvent the systemic adverse effects. Furthermore, the plasma half-life of adenosine was reported to be extremely short, less than 10 s with intravenous administration. Taken together, it is not likely that the topically applied adenosine or cordycepin elicits significant systemic adverse effects.

In this study, we have found that the adenosine and cordycepin stimulated Wnt/β-catenin pathway through adenosine receptor activation. The stimulation of adenosine receptor pathways was confirmed by elevated intracellular cAMP concentration (Figure 3a). The metabolite ligands including adenosine are effective reporters of microenvironment and are also good stimulators of regeneration mechanisms and high energy consuming processes, such as development and tissue recovery. Furthermore, we have found that not only adenosine but also cordycepin stimulated cell migration in an in vitro wound healing model and promoted cytoskeleton rearrangement, demonstrating the enhanced cell mobility of dermal fibroblasts (Figure 9).

Use of adenosine receptor agonists for the purpose of the skin wound healing has been proposed [7]. Skin aging in the field of cosmetics will be another therapeutic target for skin remodeling process since the dermal components are considered to be damaged or wounded in skin aging. Therefore, the dermal extracellular matrix synthesis and remodeling, as in wound repair processes, would improve the aged skin. In this context, cordycepin as well as adenosine, as adenosine receptor agonists, could be potential therapeutic candidates for treating skin wounds and also for aged skin as a cosmeceutical.

Our data strongly demonstrate that cordycepin, as well as adenosine, activated the Wnt/β-catenin signaling pathway and promoted cell migration and growth factor production through adenosine receptor activation. Inactivating the activity of Gsk3b via phosphorylation on Ser9 site by PKA emerged as a key step in connecting the Wnt and adenosine receptor pathways.

In conclusion, our findings provide evidences for the therapeutic potential of adenosine and cordycepin, facilitating the tissue remodeling process through the activation of adenosine receptors followed by Wnt signaling stimulation.

## 4. Materials and Methods

### 4.1. Human Dermal Fibroblast Culture

Human fibroblasts were obtained from The Department of Dermatology, Chungnam National University School of Medicine (Deajeon, Korea). Cells were cultured in DMEM (Thermofisher Scientific, Waltham, MA, USA) supplemented with 10% FBS (Thermofisher Scientific, Waltham, MA, USA). Cells were maintained in humidified incubator at 37 °C, 5% CO_2_. Before adenosine and cordycepin (Sigma-Aldrich, St. Louis, MO, USA; Figure 1a) treatment, serum limitation was done by replacing the medium with fresh DMEM supplemented with 1% FBS and culturing for 24 h to minimize the effects of serum and growth supplements. Adenosine A_2A_ receptor inhibitor ZM421385, Adenosine A_2B_ inhibitor PSB603, and PKA inhibitor H89 were purchased from Tocris Bioscience (Bristol, UK).

### 4.2. Wnt Reporter Assay

WRHEK293A cells (Amsbio, Abingdon, UK) were seeded in black 96 well plates and cultured for 24 h. Cells were treated with various concentrations of chemicals and incubated for another 24 h. Cells were then lysed by adding 50 μL of 1× Passive Lysis Buffer (Promega, Madison, WI, USA) to each well and shaking for 10 min. The expression of GFP (internal cell viability control) was assessed by measuring the fluorescence at 488/510 nm wavelength using VICTOR3 (PerkinElmer, Waltham, MA, USA). Then, 50 μL of luciferase substrate solution (Promega, Madison, WI, USA) was added to each well and the luciferase activity was measured using VICTOR3. Luminescence (TCF/LEF activity) values were normalized to GFP (cell viability) values.

### 4.3. Preparation of Cordyceps Militaris Extract

The dried *Cordyceps Militaris*, purchased from Humanherb (Daegu, Korea), were extracted with distilled water (1:15) for 1 h at 121 °C, filtered through Whatman No. 4 filter paper. The filtrate was concentrated by rotary evaporator under reduced pressure. The extract was reconstituted in water.

### 4.4. Mitochondrial Membrane Potential

The viability of fibroblast was examined using JC-1 mitochondrial membrane potential assay (Abcam, Cambridge, UK) kits following the manufacturer’s protocols. Briefly, after adenosine and cordycepin treated-fibroblasts were stained with 1 μM JC-1 solution, fluorescence intensities from JC-1 aggregate and monomer forms were measured at 590 nm (535 nm excitation) and 530 nm (475 nm excitation), respectively, with Wallac Victor3 1420 (PerkinElmer, Waltham, MA, USA). Mitochondrial membrane potential (∆Ψ) was visualized by taking fluorescence images with EVOS^TM^ FL Auto2 Imaging System (Thermofisher Scientific, Waltham, MA, USA).

### 4.5. Intracellular cAMP Measurement

The intracellular cAMP levels in cultured human dermal fibroblasts were measured using cAMP assay kit (Abcam, Cambridge, UK) following the manufacturer’s protocols. Briefly, fibroblast cells were treated with adenosine and cordycepin for 2 min and harvested. Absorbance at 450 nm was measured using microplate reader (BioTek, Winooski, VT, USA). Background wavelength correction was done at 540 nm.

### 4.6. Quantitative Real-Time PCR

Adenosine, cordycepin, and NECA were treated for appropriate times. Non-treated cells were served as negative control, while recombinant Wnt3a as a positive control for Wnt/β-catenin signaling. Total RNA was extracted using Rneasy RNA extraction kit (Qiagen Inc., Germantown, MD, USA). cDNA synthesis was performed using cDNA synthesis kit (Phillkorea, Seoul, Korea) with ThermoCycler (R&D systems, Minneapolis, MN, USA), according to the manufacturer’s protocol. cDNA samples obtained from control and treated cells were subjected to real-time (RT) PCR analysis.

TaqMan probes for RT-PCR used in this study were as follows: GAPDH assay id 4352934E; CREB1 assay id Hs00231713_m1; GSK3B assay id Hs01047719_m1; MYC assay id Hs00153408_m1, MKI67 assay id Hs04260396_g1; CCND1 assay id Hs00765553_m1; BMP2 assay id Hs00154192_m1; BMP4 assay id Hs03676628_s1; FZD3 assay id Hs00907280_m1; FZD5 assay id Hs00258278_s1, LEF1 assay id Hs01547250_m1; CTNNB1 assay id Hs00355045_m1; EGF assay id Hs01099990_m1; IGF-1 assay id Hs01547656_m1; TGFB1 assay id Hs00998133_m1; TGFB2 assay id Hs00234244_m1; VEGFA assay id Hs00900055_m1.

TaqMan One-Step RT-PCR Master Mix Reagent (Life Technologies, Carlsbad, CA, USA) was used. The PCR reactions were performed on ABI 7500 Real Time PCR system following the manufacturer’s instruction. The resulting data were analyzed with ABI software.

### 4.7. Western Blot Analysis

Fibroblast cells (1 × 10^6^ cells/dish) were seeded in 100 mm dishes and cultured for 24 h. Adenosine and cordycepin were treated for 4 h. The cells were then lysed and total cellular proteins were prepared. Then, 50 μg protein samples were analyzed by Western blotting with corresponding antibodies; GSK-3β (27C10) (1:1000, Cell Signaling Technology, Danvers, MA, USA), Phospho-GSK-3β (Ser9) (Cell Signaling Technology) GAPDH (1:2000, Santa Cruz, CA, USA). Western blot was analyzed by chemiluminescence detector iBright FL1000 (Invitrogen, Waltham, MA, USA).

### 4.8. Protein Dot Blot Analysis for Growth Factors (Receptors) and MAP Kinase Phosphorylation

Human growth factor antibody array kit (Abcam, Cambridge, UK) and human MAP kinase phosphorylation antibody array kit (Abcam, Cambridge, UK) were used to elucidate the changes in growth factor profiles and signal transduction pathways in cultured human dermal fibroblasts. A total of 41 human growth factors and 17 human MAPK phosphorylation were analyzed. Briefly, cells were treated with 1.5, 3 mM of adenosine and 50, 100 µM of cordycepin for appropriate time and then collected for growth factor and MAPK phosphorylation analysis. Cells treated with vehicle medium were used as non-treated control. Conventional immunoblot process was performed following the manufacturer’s protocol. The resulting blots were analyzed under identical condition using iBright FL1000 (Invitrogen, Waltham, MA, USA).

### 4.9. In-Vitro Scratch Wound Healing Assays

Fibroblast cells (2 × 10^5^ cells) were seeded in Culture Insert-2 Well (ibidi GmbH, Munich, Germany) and cultured for 24 h. After the culture inserts were removed, medium was replaced with DMEM containing various concentrations of adenosine, cordycepin and wnt3a and incubated for two days. Images with time intervals were obtained using EVOS^TM^ FL Auto2 Imaging System (Thermofisher Scientific, Waltham, MA, USA). Cell migration rates were calculated using the image segmentation algorithm on Image J plugin, described elsewhere [51].

### 4.10. Immunocytochemistry

Fibroblast cells (8 × 10^4^ cells per well) were seeded in 24 well plates and cultured overnight. After PBS wash, cells were fixed with 4% paraformaldehyde at room temperature for 10 min. Cells were then permeabilized with PBS containing 0.1% triton x-100 and blocked with PBS containing 5% FBS and 1% BSA. After consecutive incubation with primary antibodies (1:200 dilution, Abcam, Cambridge, UK) at 4 °C for 12 h and alexa 488 nm or alexa 594 nm conjugated secondary antibodies (1:1000 dilution, Thermofisher Scientific, Waltham, MA, USA) at room temperature for 1 h, nuclei were stained with DAPI (1:2000 dilution, Thermofisher Scientific, Waltham, MA, USA) in the dark for 10 min. High resolution fluorescence images were taken using EVOSTM FL Auto2 Imaging System (Thermofisher Scientific, Waltham, MA, USA).

### 4.11. Statistical Analysis

All experimental data were presented as the mean ± standard deviation (S.D.) of at least three independent experiments. Experimental results were analyzed using the SigmaPlot 8.0 (Systat Software Inc., Chicago, IL, USA). The statistical significance of difference was determined by Student’s *t*-test. The value of *p* < 0.05 was considered statistically significant.

## Figures and Tables

**Figure 1 ijms-22-05571-f001:**
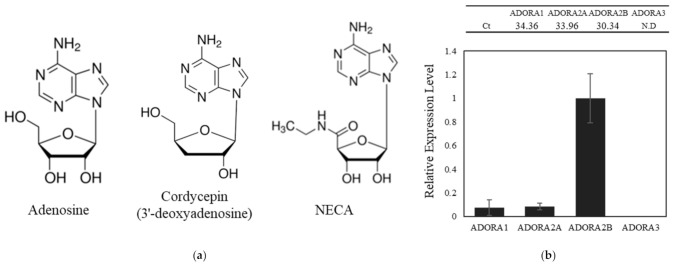
Chemical structures of adenosine, cordycepin, and NECA and the mRNA expression profile of adenosine receptor subtypes. (**a**) Chemical structures. (**b**) The expression of four adenosine receptor subtypes, A_1_ (ADORA1), A_2A_ (ADORA2A), A_2B_ (ADORA2B), and A_3_ (ADORA3) was examined using quantitative RT-PCR analysis in cultured human dermal fibroblast. Ct values for each receptor subtype were indicated.

**Figure 2 ijms-22-05571-f002:**
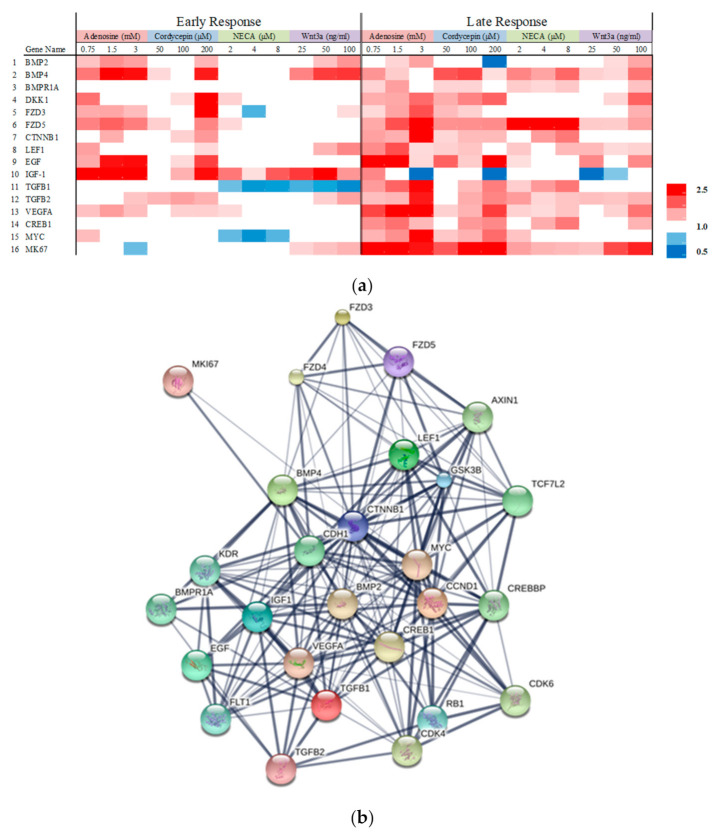
The changes in mRNA expression profiles by adenosine analogues and recombinant wnt3a in cultured human dermal fibroblasts. Cells were treated with adenosine, cordycepin, NECA, and wnt3a for one day (early response) and two days (late response) and mRNA expression was examined by RT-PCR. (**a**) Sixteen genes with more than a two-fold increase and statistical significance (*p* < 0.05) was displayed, red and blue color mean increase and decrease in the mRNA expression, respectively, (**b**) further investigated by protein cluster analysis.

**Figure 3 ijms-22-05571-f003:**
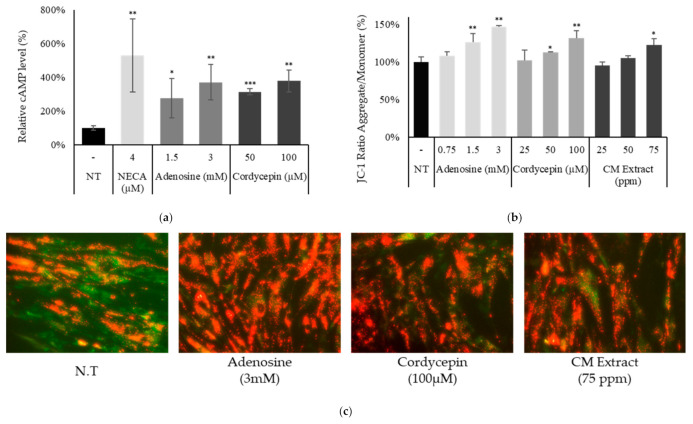
Adenosine and cordycepin activated adenosine receptor pathway in cultured human dermal fibroblast. Cells were treated with adenosine, cordycepin, NECA, and CM extract. (**a**) After 4 h treatment, intracellular cAMP levels were measured. NECA was used as a positive control for adenosine receptor activation. (**b**) Mitochondrial membrane potential was measured after 4 h treatment with adenosine, cordycepin, and CM extract. (**c**) Representative photos of JC-1 staining. JC-1 monomer form was seen as green and aggregate form as red by fluorescent microscopy. (**d**) The mRNA expression levels of CREB1, GSK3B, MYC, Ki67, and CCND1 were measured after treatment with adenosine and cordycepin for one day and two days. N.T, non-treated control; Significantly different compared with N.T (* *p* < 0.05, ** *p* < 0.01, *** *p* < 0.001).

**Figure 4 ijms-22-05571-f004:**
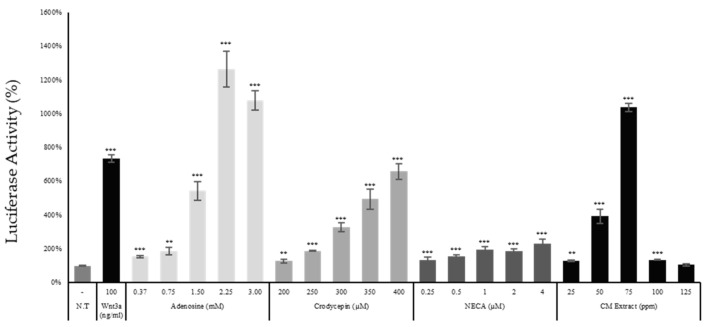
Adenosine, cordycepin, and NECA stimulated Wnt reporter activity. The Wnt reporter WRHEK293A cells were treated with adenosine, cordycepin, NECA, and CM extract and the luciferase activity was measured. N.T, non-treated control; significantly different compared with N.T (* *p* < 0.05, ** *p* < 0.01, *** *p* < 0.001).

**Figure 5 ijms-22-05571-f005:**
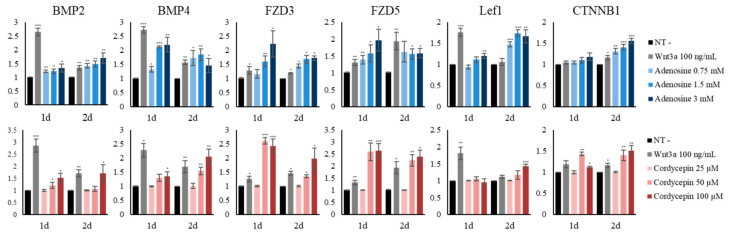
Adenosine and cordycepin activated Wnt/β-catenin signaling pathway in cultured human fibroblasts. The expression levels of 6 Wnt/β-catenin pathway related genes (BMP2/4, FZD3/5, LEF1 and CTNNB1) were assessed in fibroblasts treated with adenosine and cordycepin for one day and two days. Recombinant wnt3a was used as a positive control. The data represent the means of six independent samples. N.T, non-treated control; significantly different compared with N.T (* *p* < 0.05, ** *p* < 0.01, *** *p* < 0.001).

**Figure 6 ijms-22-05571-f006:**
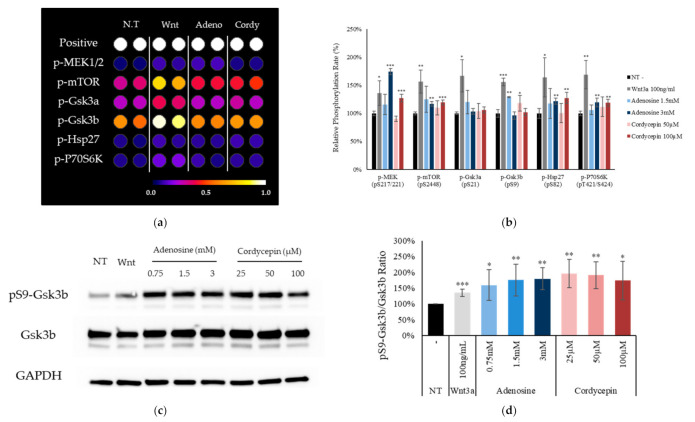
Adenosine and cordycepin increased phosphorylation of Gsk3b on Ser9 in cultured human fibroblasts. (**a**,**b**) Cells were treated with adenosine and cordycepin for one day. Whole cell lysates were analyzed by immunoblotting to determine the phosphorylation levels of MAP kinases following manufacturer’s instruction. A total of 17 MAP kinases were analyzed. Six kinases with significantly increased phosphorylation were displayed. (**c**) The levels of phospho-Gsk3b, Gsk3b were further investigated by Western blotting after 4 h treatment with adenosine and cordycepin. (**d**) The ratio of p-Gsk3b/Gsk3b was calculated. The data represent the means of six independent samples. N.T, non-treated control; significantly different compared with N.T (* *p* < 0.05, ** *p* < 0.01, *** *p* < 0.001).

**Figure 7 ijms-22-05571-f007:**
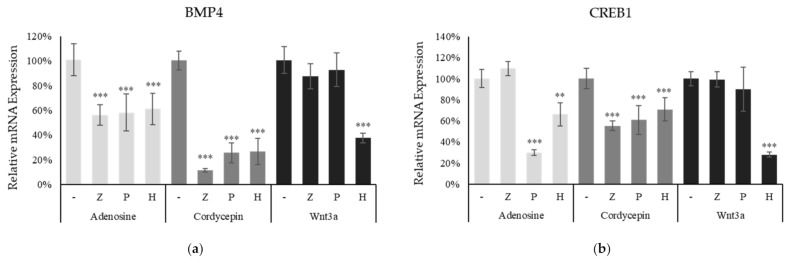
Activation of Wnt/β-catenin pathway by adenosine and cordycepin was abolished in the presence of adenosine receptor antagonists and PKA inhibitor. The cells were treated with adenosine and cordycepin in the presence of various inhibitors (ZM241385, PSB603 and H89) for 24 h, harvested, and the mRNA expression levels of BMP4 (**a**) and CREB1 (**b**) were measured by RT-PCR. The data represent the means of six independent samples. Significantly different compared with each single adenosine, cordycepin and wnt3a treatment. (** *p* < 0.01, *** *p* < 0.001); Adenosine (3 mM); Cordycepin (100 µM); Wnt3a (100 ng/mL); Z: ZM241385 (20 µM); P: PSB603 (10 µM); H: H89 (10 µM).

**Figure 8 ijms-22-05571-f008:**
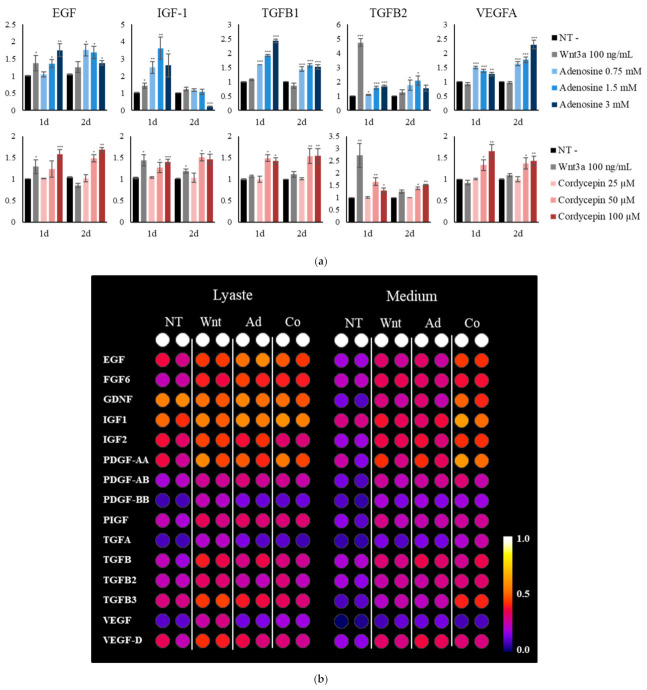
Adenosine and cordycepin increased the production of growth factors associated with the tissue repair process. The mRNA and protein expression levels of growth factors were examined by RT-PCR and dot blot analysis, respectively. (**a**) Human dermal fibroblast cells were treated with adenosine and cordycepin for one day and two days. Recombinant wnt3a and vehicle medium served as positive and non-treated control, respectively. The mRNA expression levels of five growth factor genes were measured. (**b**) The cells were treated with adenosine and cordycepin for 24 h, and then medium was collected, assessed by dot blot analysis. Results of 15 growth factors were displayed. (**c**) The band intensities were quantitated. Positive, biotin-conjugated IgG. The data represent the means of four independent samples. Significantly different compared with N.T (* *p* < 0.05, ** *p* < 0.01, *** *p* < 0.001).

**Figure 9 ijms-22-05571-f009:**
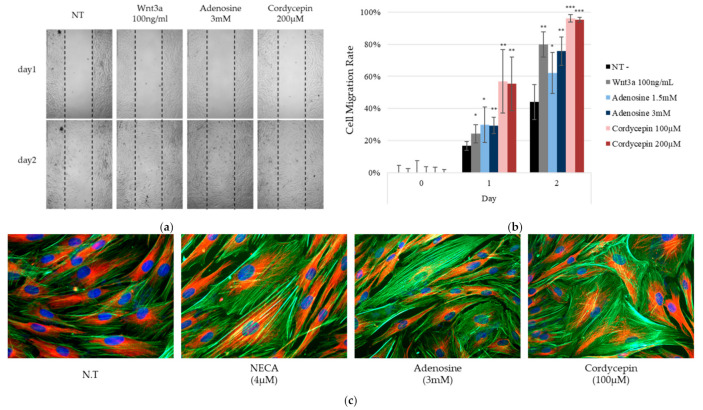
Adenosine and cordycepin promote the in vitro tissue remodeling process and activation of the adenosine receptor resulted in cytoskeletal rearrangement. In order to investigate the possible role of adenosine and cordycepin in the wound healing process, the cell migration of fibroblasts was evaluated by cell scratch assay in the presence of adenosine, cordycepin, and Wnt3a. (**a**) At day one and two, the cells were photo-documented. (**b**) The migration rates were analyzed. Wnt3a was used as a positive control. (**c**) Rearrangement of cytoskeleton was visualized with β-tubulin (Red) and F-actin (Phalloidin, Green). Nuclei were stained with DAPI (blue). NECA was used as a positive control for the adenosine receptor agonist. The data represent the means of four independent samples. Significantly different compared with N.T (* *p* < 0.05, ** *p* < 0.01, *** *p* < 0.001).

**Table 1 ijms-22-05571-t001:** GO analysis for mRNA expression profile changes by adenosine analogues.

No.	GO Pathway ID	Pathway Description	Gene Count	*p*-Value
1	GO.0050678	Regulation of epithelial cell proliferation	15	5.09 × 10^−18^
2	GO.0051240	Positive regulation of multicellular organismal process	21	2.91 × 10^−17^
3	GO.0010604	Positive regulation of macromolecule metabolic process	24	3.60 × 10^−16^
4	GO.0010557	Positive regulation of macromolecule biosynthetic process	21	5.45 × 10^−16^
5	GO.0010628	Positive regulation of gene expression	21	9.98 × 10^−16^
6	GO.0048522	Positive regulation of cellular process	26	7.49 × 10^−16^
7	GO.0048468	Cell development	20	1.12 × 10^−16^
8	GO.0051094	Positive regulation of developmental process	18	1.28 × 10^−14^

**Table 2 ijms-22-05571-t002:** KEGG analysis for mRNA expression profile changes by adenosine analogues.

No.	KEGG Pathway ID	Pathway Description	Gene Count	*p*-Value
1	4390	Hippo signaling pathway	16	3.40 × 10^−26^
2	4310	Wnt signaling pathway	11	3.34 × 10^−16^
3	4110	Cell cycle	9	6.62 × 10^−13^
4	4151	PI3K-Akt signaling pathway	11	3.93 × 10^−12^
5	4350	TGF-beta signaling pathway	7	1.02 × 10^−10^

**Table 3 ijms-22-05571-t003:** Responses of the inhibitors to adenosine, cordycepin, and Wnt3a induction.

Inhibitor Name	Target	Response to Ligands ^1^		
		Adenosine	Cordycepin	Wnt3a
ZM241385	A_2A_ Receptor	△	O	X
PSB603	A_2B_ Receptor	O	O	X
H89	PKA	O	O	O

^1^ △: not certain; O: inhibited; X: not inhibited.

## Data Availability

The data presented in this study are available on request from the corresponding author. The data are not publicly available due to the policy of company.

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
