# Peer review of "Adenosine and Cordycepin Accelerate Tissue Remodeling Process through Adenosine Receptor Mediated Wnt/β-Catenin Pathway Stimulation by Regulating GSK3b Activity"

_ijms, 2021, doi:10.3390/ijms22115571_

Round 1
Reviewer 1 Report
The authors present a paper about the effects of Adenosine-like agents on the expression of wound healing factors on fibroblasts. I find the paper well-done but I have a couple of questions.
1) The authors study only one cell type - fibroblasts. Do they feel that there would be similar findings for keratinocytes or endothelial cells, cells that are also important for wound healing?
2) Do the authors feel that if fibroblasts were co-cultured with keratinocytes that there would be similar expression changes (ie. skin healing)?
3) Adenosine, given systemically has profound effects on the body - especially the heart. How do the authors feel that adenosine or the other similar agents would be applied to wounds without significant systemic effects?
Author Response
Point 1 : The authors study only one cell type - fibroblasts. Do they feel that there would be similar findings for keratinocytes or endothelial cells, cells that are also important for wound healing?
Response 1 : As you mentioned, other cell types including keratinocyte and endothelial cells play important roles in wound healing process. The Wnt/β-catenin also plays pivotal roles in epidermal regeneration and angiogenesis, which are important in wound healing process. We don’t know, however, adenosine would exert similar effects in these cell types through the same molecular mechanisms since we don’t have any evidences. Although our study focused only on the fibroblast cells because of its major role in wound repair, we think those will be very interesting research themes.
Point 2 : Do the authors feel that if fibroblasts were co-cultured with keratinocytes that there would be similar expression changes (ie. skin healing)?
Response 2 : Apart from the direct effects of adenosine on keratinocytes which are unknown yet, several cytokines including EGF, IGF-1, TGFB1/2, and VEGFA were stimulated by adenosine. In co-culture system we can expect the effects of secreted cytokines on the keratinocyte, one of which would be epidermal (keratinocyte) proliferation.
Point 3 : Adenosine, given systemically has profound effects on the body - especially the heart. How do the authors feel that adenosine or the other similar agents would be applied to wounds without significant systemic effects?
Response 3 : We focused on skin wounds which are mostly external body sites and are target for topical application. Although topically applied chemicals could enter the circulation, we think topical application would be a good regimen to circumvent the systemic adverse effects. Furthemore, the plasma half-life of adenosine was reported to be extremely short (less than 10 seconds with intravenous doses).
Taken together, we don’t think that topically applied adenosine likely elicit a significant systemic adverse effects.

Reviewer 2 Report
The ms needs only few minor revisions. I suggest to check better the text to avoid some english errors. I also suggest to cut the discussion sections, to put more emphasis on the obtained results as well as on the possible applications.
Author Response
Point 1 : The ms needs only few minor revisions. I suggest to check better the text to avoid some English errors. I also suggest to cut the discussion sections, to put more emphasis on the obtained results as well as on the possible applications.
Response 1 : As you suggested, we corrected English errors and deleted some parts of the discussion section seemed not necessary.
